# Impact of High-Pressure Processing on Antioxidant Activity during Storage of Fruits and Fruit Products: A Review

**DOI:** 10.3390/molecules26175265

**Published:** 2021-08-30

**Authors:** Concepción Pérez-Lamela, Inmaculada Franco, Elena Falqué

**Affiliations:** 1Nutrition and Bromatology Group, Department of Analytical and Food Chemistry, Faculty of Sciences, University of Vigo—Ourense Campus, E32004 Ourense, Spain; 2Food Technology Area, Faculty of Sciences, University of Vigo—Ourense Campus, E32004 Ourense, Spain; inmatec@uvigo.es; 3Analytical Chemistry Group, Department of Analytical and Food Chemistry, Faculty of Sciences, University of Vigo–Ourense Campus, E32004 Ourense, Spain; efalque@uvigo.es

**Keywords:** high hydrostatic pressure, refrigerated or ambient storage, fruit preparations, antioxidant capacity

## Abstract

Fruits and fruit products are an essential part of the human diet. Their health benefits are directly related to their content of valuable bioactive compounds, such as polyphenols, anthocyanins, or vitamins. Heat treatments allow the production of stable and safe products; however, their sensory quality and chemical composition are subject to significant negative changes. The use of emerging non-thermal technologies, such as HPP (High Pressure Processing), has the potential to inactivate the microbial load while exerting minimal effects on the nutritional and organoleptic properties of food products. HPP is an adequate alternative to heat treatments and simultaneously achieves the purposes of preservation and maintenance of freshness characteristics and health benefits of the final products. However, compounds responsible for antioxidant activity can be significantly affected during treatment and storage of HPP-processed products. Therefore, this article reviews the effect of HPP treatment and subsequent storage on the antioxidant activity (oxygen radical absorbance capacity (ORAC) assay), 2,2-diphenyl-1-picrylhydrazyl (DPPH) radical scavenging capacity assay, ferric reducing antioxidant power (FRAP) assay, 2,2′-azino-bis-(3-ethylbenzothiazoline-6-sulfonic acid) (ABTS) radical scavenging capacity assay or Trolox equivalent antioxidant capacity (TEAC) assay), and on the total phenolic, flavonoid, carotenoid, anthocyanin and vitamin contents of fruits and different processed fruit-based products.

## 1. Introduction

Fruits are very important to maintain a healthy diet due to the content of antioxidant compounds such as vitamins (ascorbic acid and tocopherols), minerals (selenium), polyphenols (flavonoids and anthocyanins) and other components such as fiber (pectins and cellulose). Moreover, other substances give flavor and attractive colors to enhance their sensory properties. It is widely known that fruits and vegetables are outstanding sources of phytochemicals with biological activity. Their activity has been related to the scavenging of free radicals and nonradical reactive oxygen species [1] contributing to their antioxidant power. All of these properties, together with the World Health Organization (WHO) dietary recommendations (intake of 400 g/day of fruits and vegetables or 5 portions) make these products attractive to be consumed.

Nevertheless, fruits are seasonal and highly perishable, containing more than 80% water, which make them undergo progressive changes if stored untreated at ambient temperature due to the activity and spoilage caused by microorganisms, molds, insects or to naturally present enzymes such as AAO (ascorbic acid oxidase), PPO (polyphenoloxidase), and POD (peroxidase). These enzymes are known to cause quality modifications in texture, flavor, color, and nutritive value [2,3].

Some authors have proposed intervention programs to promote fruit and vegetable consumption [4,5,6] considering the association of their low consumption with some diseases, pointed out from the beginning of 2000s [7,8]. Most of them are called non-communicable diseases by WHO [9]: cancers [10,11,12], obesity [13], high blood pressure [14] and other cardiovascular illnesses [15], strokes [16], osteoporosis [17], osteoarthritis [18], type 2 diabetes [19], degenerative diseases [20], among others. They are known to reduce the risk of cancer, heart disease, and diabetes; to inhibit plasma platelet aggregation, cyclooxygenase activity, and histamine release, as well as to exert antibacterial, antiviral, anti-inflammatory, and anti-allergenic activities [21].

The food industry produces a huge amount of fruit and vegetable residues or by-products every year around the globe from many different sources [22]. Food waste is a worldwide concern as it represents a constant threat to the environment and a serious operational problem for the food industry [23]. It is estimated that food waste for fruits/vegetables is around 0.5 billion tons per year of the 1.3 billion tons of total waste [24]. Nearly half of the horticultural produce (fruits, vegetables, and root crops) is wasted globally, reaching up to 60% [25]. Some authors report that major post-harvest losses (30–40%) occur in fruits and vegetables [26]. Food wastes imply significant greenhouse gas emissions that increase the challenge of climate change and impact food security [27]. HPP can use surplus and discarded fruits to be processed, contributing to minimize fruits/vegetables losses up to 15% in some fruits [28], allowing the industries to achieve the concept of zero-waste economy and sustainable society [29], avoiding economic loss, and contributing to a circular economy. 

In the last decades, the market has evolved considerably due to the development of numerous technologies for food preservation. The main factor for choosing non-thermal food processing technology is the improvement of nutritional and sensory properties [30]. In order to extend the shelf-life and to preserve different fruit products, high pressure processing is considered one of the most widely used emerging non-thermal technologies in recent years. The most impressive advantage of HPP is the potential to inactivate enzymes and microbial load, while exerting rather minimal effects on nutritional and organoleptic properties of food products. 

Several authors have pointed out that the evidence for specific health benefits attributed to the antioxidant content of foods was still limited 10 years ago [31]. Additionally, the processing of vegetable food products and their subsequent storage conditions may have a positive or negative influence on the stability of compounds [32]. Therefore, the current review intends to provide an overview of antioxidant activity and antioxidant compounds affected by HPP in fruits and fruit products after their storage (at 4 °C–25 °C) from 1 week to 6 months (Figure 1). This review focused on the latest knowledge on the effects of this processing technology on antioxidant activity from 2015.

## 2. Fruits and Their Antioxidant Activity

### 2.1. Determination of Antioxidant Activity/Capacity

Fruits are excellent sources of antioxidants [33] that protect cells against the damaging effect of reactive oxygen species (ROS) and neutralize or interfere the mechanisms of the reactive free radicals, such as hydrogen peroxide, superoxide radical, and hydroxyl radical, preventing or reducing the amount of damage. Their antioxidant ability is related to their chemical composition; mainly to the presence of phenolic compounds (flavonoids and phenolic acids) and other biologically active compounds, including vitamins (C, B, A, and E), carotenoids, minerals (Zn, Se), etc. [34,35]. These natural antioxidants possess some biological functions, such as anti-inflammatory, anti-cancerous, anti-degenerative, anti-aging, and anti-diabetes properties, among others [36,37,38].

This antioxidant function has been labelled with some terms, such as antioxidant activity, antioxidant capacity, antioxidant power, antioxidant properties, antioxidant potential, antioxidant profile, antioxidant content, antioxidant components, antioxidant composition, etc. The two first mentioned are the most common, although they are not synonymous. According to Apak [39], antioxidant capacity “has a thermodynamic definition concerning the oxidative conversion efficiency (linked to stoichiometry) of an antioxidant by an oxidizing agent and is related to the equilibrium constant of this conversion… and the units is generally the number of moles of reactive species scavenged by 1 mol of antioxidant during a fixed time period” and antioxidant activity “is mainly concerned with reaction kinetics, in that it reflects the rate of chemical oxidation of the tested antioxidant or the rate of quenching of a given reactive species by the antioxidant… and the units are usually lag times, percentage inhibition or scavenging relative to a reference compound, and final reaction rate constants”. In addition, the concept of antioxidant, as well as their implications, differ in food and biomedical sciences and the antioxidant indexes obtained by chemical assays frequently cannot extrapolate the study *in vivo* [40,41].

A large number of analytical methods, usually employing luminescence or visible spectroscopic techniques, have been employed for determination of antioxidant capacity of foods. Numerous reviews describe this large variety of methods, classifications, assay descriptions, usefulness and limitations, advantages, and disadvantages [35,39,42,43].

In general, *in vitro* methods for determining antioxidant properties can be divided based on reaction mechanisms into: hydrogen atom transfer (HAT), electron transfer (ET) or single electron transfer (SET), and mixed mode (HAT- and ET-based) methods [39,42]. The end result is the same, regardless of the mechanism, but kinetics and potential for side reactions are different. Figure 2 shows the characteristics of these reaction mechanisms and the main assays based on them. Other authors [44] proposed another classification, dividing the antioxidant assays into two groups: methods based on radical/ROS scavenging (such as ORAC, Chemiluminescence assay, DPPH, TEAC, etc.) and based on non-radical redox potential (such as FRAP, CUPRAC, CERAC, CHROMAC, etc.).

Other analytical techniques used for determination of the antioxidant activity or capacity are [45]: (i) Electrochemical techniques, such as cyclic voltammetry, based on the inversion of the working electrode’s potential ramp when the value of a set potential is reached, the amperometric and biamperometric methods, both based on the reduction of 2,2-diphenyl-1-picrylhydrazyl (DPPH^•^); (ii) Biosensors methods based on monitoring of superoxide radical, monitoring of nitric oxide, monitoring of glutathione, monitoring of uric acid, ascorbic acid or phenolic compounds; (iii) Chromatographic methods based on the separation, detection, and quantification of individual compounds, HPLC (high performance liquid chromatography), usually with a diode array detector (DAD), being the most employed. Recently other methods have been optimized in food matrices for antioxidant analysis, such as synchronous fluorescence spectroscopy [46], electrochemical method based on *in situ* formation of free superoxide radicals [47], and electron paramagnetic resonance spectrometry-DPPH assay [48].

The main chemical *in vitro* methods used for the determination of the antioxidant properties in fruits and fruit-based products are the following (Figure 3) [34,35,45]:

Oxygen Radical Absorbance Capacity (ORAC) assay measures the antioxidant scavenging activity against the peroxyl radical by HAT-based mechanism. The radicals, generated from the thermal decomposition of 2,2′-azo bis (2-amidino-propane) dihydrochloride in aqueous buffer, reacts with a fluorescent molecule, such as fluorescein, and measure the fluorescence (excitation at 485 nm and emission at 520 nm, 528 nm, or 535 nm) decrease of the non-fluorescent product formed. ORAC values are usually reported as Trolox equivalents. Three samples collected in this review analyzed antioxidant activity with this method and expressed values in µmol Trolox equivalent/100 mL in a juice concentrated, in μmol Trolox equivalent/100 g fresh weight in cantaloupe purée, Trolox equivalent mM in a tiger nuts’ milk.

Ferric Reducing Antioxidant Power (FRAP) assay is based on the reduction of the 4,6-tri(2-pyridyl)-1,3,5-triazine (TPTZ) by the presence of antioxidants by SET-mechanism. When the colorless complex ferric ion-TPTZ is reduced, an intense navy blue colored ferrous complex is formed and the absorbance can be measured at 593 nm. Results are usually expressed as micromolar Fe^2+^ equivalents or relative to an antioxidant standard (Trolox or ascorbic acid). The studies revised in this review have showed this value in mmol FeSO_4_/g (in apple juice), mmol of Fe^2+^ equivalents/mL (in aronia or maoberry juices), μmol Trolox equivalents/g fruit in dry weight (in tomato juice), mmol Trolox equivalents/L (in mulberry, Korla pear or red grapefruit juices), mg Trolox/100 g flesh (in cupped strawberry or strawberry flesh in syrup), μmol Trolox/kg smoothie or 100 g smoothie, μmol Fe^2+^/L or /100 mL in smoothie samples.

Total phenolic content (TPC) or Folin–Ciocalteu reducing capacity (FCR assay) is actually not an antioxidant method, however, phenolic compounds are associated with high antioxidant ability and the basic mechanism is an oxidation/reduction reaction and, therefore, it can be considered as another antioxidant method. Furthermore, it is precise, sensitive, and simple, and it is widely used to characterize the total phenolic content of foods. Folin–Ciocalteu reagent contain phosphomolybdic/phosphotungstic acid complexes. The transfer of electrons (SET mechanism) from phenolic compounds and other reducing species to this reagent yield a blue colored complexes with maximum absorbance at 750–765 nm. Frequently it is determined as gallic acid or another phenolic compound (caffeic acid, catechin, chlorogenic acid, or ferulic acid) equivalent. The 88% of the studies collected in this review analyzed TPC and the results were expressed mainly as mg or g of gallic acid equivalent (GAE) related to liter, 100 mL, 100 g or kg of sample, 100 g fresh weight or g in dry based; one sample was referred to mg Chlorogenic acid/kg, two samples to mg Catechin equivalents/kg or /mL and another in mg Rutin equivalents/100 g of fresh weight.

2,2-diphenyl-1-picrylhydrazyl radical (DPPH^•^) scavenging assay is included in mixed mode (HAT- and SET-based) methods, although the predominant mechanism is SET. This assay is widely used to evaluate the free radical scavenging potentials of different compounds. The solutions of DPPH radical have an intense purple color, which absorb at 515–520 nm, and decrease when reacting with antioxidant compounds (containing CH, NH, OH, and SH groups in its molecule) forming the reduced form of DPPH^•^ (DPPH hydrazine) with a pale yellow color. Results are usually referred to a standard antioxidant, such as Trolox, or as the antioxidant concentration that provides 50% inhibition of the DPPH radical (labelled as IC50); both units were mainly used in all works reviewed, except in fresh mango and jujube pulp (Table 1) that were expressed as g vitamin C equivalent/kg of fresh weight and mg ascorbic acid/100 g of fresh weight, respectively.

TEAC (Trolox equivalent antioxidant capacity) or other ABTS (2,2′-azino-bis-(3-ethylbenzothiazoline-6-sulfonic acid) radical) assays are also considered with a mixed mode (HAT- and SET-based) mechanism. Solutions of ABTS cation radical (ABTS^•+^) have a blue-green color with maximum absorbance at wavelengths of 415, 645, and 734, and 815 nm (415 and 734 nm are the most used), that decolorize in the presence of peroxyl radicals or other oxidants. As for the DPPH method, results were frequently expressed referring to Trolox, or as IC50 (50% inhibition). This method was only applied in seven samples (Tables 1 and 2) and expressed as Trolox and vitamin C equivalents, and in one sample referred as % antiradical activity.

Total vitamins, carotenes, flavonoids, and anthocyanins contents were also included in Tables 1–3 because these families of compounds are characterized by a higher antioxidant capacity [35]. The analyses usually were carried out by spectroscopy at different wavelengths. Even, some works use the separation by HPLC-DAD). In addition, several authors found a positive correlation between some of these compounds or families and the antioxidant activity.

### 2.2. Vitamins and Phenolic Compounds Related to Antioxidant Capacity

Fruits and vegetables, being rich in phenolic compounds (flavonoids, anthocyanins, phenolic acids, among others), carotenoids, glucosinolates, vitamin C, and tocopherols, are plentiful in bioactive compounds [49,50,51]. Due to their high antioxidant capacity, they are important for preventing oxidative stress and chronic diseases. The clinical significance of the beneficial properties of plant polyphenols was first reported in 1950 [52]. In addition, inverse associations between fruit and vegetable intake and chronic diseases, such as different types of cancer and cardiovascular disease, have been demonstrated in numerous epidemiological studies [53].

The bioactive compounds from the plant secondary metabolism have a clear therapeutic potential by contributing towards the antioxidant activities. Various authors re-ported that the phenolic compounds, vitamin C, and carotenoids are the main phytochemicals present in fruits and vegetables that are related to human health [54]. Main antioxidants in vegetable food include some other constituents such as enzymes, vitamins, minerals, and diverse groups of phytochemicals such as polyphenols, allylsulfides, sulfur-containing compounds, betalains, saponins, phytosterols, and capsaicinoids [32].

Phenolic compounds act as primary antioxidants or free radical terminators, with their antioxidant activities based on their ability to donate hydrogen atoms to free radicals. Anthocyanins, a type of phenolic compounds, present in fruits and vegetables with pink to red and violet hues, have been reported to be associated with health benefits such as preventing cardiovascular diseases [55], inhibiting tumor cell [56] and protecting skin from ultraviolet radiation [57]. However, anthocyanins are prone to degradation when they are exposed to light, metal ions, and heat treatment. As a result, the antioxidant capacity, color, flavor, and nutrients of fruit-based food products would be affected adversely by improper processing methods or storage conditions [58]. Therefore, it has become an essential research focus to maintain the stability and antioxidant capability of anthocyanins in processed fruit products [59]. Flavonoids are another group of phenolic compounds predominantly present in fruits and vegetables and known to possess strong anti-inflammatory activity [60]. They also play an important role to prevent certain diseases such as neurodegenerative processes [61] and metabolic disorders [62]. Although the oral bioavailability of flavonoids in some fruit products was relatively low, these compounds displayed a significant effect on the regulation of metabolism to ameliorate metabolic disorders.

The main active forms for vitamin A are carotenoids. The potential beneficial health properties of carotenoids, in particular their anti-inflammatory and antioxidant effects, have been widely recognized and have long been considered an interesting study target [63,64]. Most investigations have traditionally focused on evaluating food carotenoid content. It should be kept in mind that the positive effect of these secondary plant compounds, or any other functional compounds, depends not only on their content but also on the extent to which they are bioaccessible and available for absorption after ingestion and digestion [65].

Vitamin C, which includes two compounds with antioxidant activity (L-ascorbic and L-dehydroascorbic acids), is the most studied vitamin, considering that it acts like an indicator of food quality. Stability of these acids depends on the amount of oxygen, oxidation being the main cause of their degradation. The stability of vitamin C during storage in fruits and fruit-based products is related to enzyme activity and the availability of oxidants that catalyze the oxidation of ascorbic acid. Ascorbic acid retention has also been related with the pH of the samples and with storage temperature since higher temperatures could accelerate the oxidation rates. The lowest storage temperature usually allows for the best vitamin C retention [66].

Vitamin E (α-tocopherol) is primarily an antioxidant contained in fruits such as avocado, mango, and kiwi. It provides effective protection against lipid peroxidation, DNA mutations, mitochondrial damage, loss of neurons, and Aβ deposition. An important function of α-tocopherol is protecting from oxidation fatty acids in membrane phospho-lipids and plasma lipoproteins [67].

Nutritional epidemiological surveys have shown that dietary vitamin E or use of combined α-tocopherol and γ-tocopherol is associated with a delay of age-related cognitive decline along with a lower risk of Alzheimer’s disease [68]. In order to recover its unoxidized state, vitamin E must interact with other antioxidants such as ascorbate (vitamin C) and glutathione. The free-radical scavenger function depends not only on the α-tocopherol content, but also on the degree of oxidative stress.

When any preservation method (as for example HPP) is employed in the processing of food, it is important that bioavailability of key components such as bioactive compounds will be maintained. The term bioavailability has several working conditions and there is no universally accepted definition. From a nutritional point of view, it is defined as the fraction of ingested component available for utilization in normal physiological functions. Some studies have reviewed works on bioavailability for various phytonutrients [69,70] and, more specifically, for vitamin C [71], carotenoids [72,73]. and other bioactive substances such as vitamin A, folates, and vitamin K [74]. Nevertheless, they are still needed for phenolic compounds in some concrete cases such as berries [75]. In other fruits, the bioavailability has recently been reviewed for apples [76] and tomatoes [77]. Finally, some authors have reviewed the effect of processing techniques on bioavailability and bioaccessibility in various bioactive compounds, reporting that, in general, HPP allows to increase both of them [78].

Nowadays, the positive association between antioxidant activity and phytonutrients has been reported by various authors [79]. Other studies also found positive correlations, between antioxidant capacity and some bioactive compounds in various fruits [80] or in concrete fruits: doum [81], guava [82], black wolfberry [83], and strawberry [84]. Less correlation was exhibited in one article studying acerola, for vitamin C and total phenolic compounds [85], although one other work found a positive correlation only for ascorbic acid, also in acerola [86]. A positive effect was found in citrus fruits for bioactive compounds and antioxidant activity measured using the DPPH method [87], but weak correlation was reported for citrus fruits between vitamin C and carotenoids, with lipophilic antioxidant capacity [88], and also between vitamin C and flavonoids in blackberries [89]. So, most of the studies in the recent years have published positive correlations that give importance to the consumption of fruits and fruit preparations for a healthy diet.

## 3. Technological Treatment of Fruits and Fruit Products

### 3.1. Conventional Methods

Traditional heat treatment methods, such as pasteurization and commercial sterilization, are widely used in the fruit processing industry for their effective preservative effect, due to the destruction of microorganisms and inactivation of enzymes. However, excessive treatment can lead to losses in nutritional quality and phytochemical content, as well as the physicochemical, rheological, and organoleptic properties of processed products [90].

Commercial sterilization is a more drastic thermal process, where the final product is microbiologically safe and guaranteed to be stable for long shelf-life. Pasteurization is a milder thermal process, whose main objectives are to ensure safety and prolong food shelf-life. It is a method that needs to be used with a complementary technology (such as cooling, acidification, reduction of water activity, and/or use of preservatives) to guarantee stability. Pasteurized products have low to medium stability and variable shelf-life, according to their characteristics and complementary conservation method [91].

Pasteurization is effective in inactivating some undesirable enzymes, such as POD and PPO, which can contribute to enzymatic browning and affect the quality of different fruit-based products. Normally, a short heat treatment at 80–100 °C is sufficient to inactivate some unwanted enzymes. Mild heat treatments (60 °C, 30 min) on pawpaw pulp achieved 70% inactivation of PPO [92]. On the other hand, some authors [93] found that when blueberry purée was treated at different temperatures (40, 50, 60, 70, 80, 90, and 100 °C) for 20 min the PPO activity decreased with increasing temperature. The activity was inactive when the processing temperature was higher than 80 °C. In addition, β-glucosidase activity was significantly decreased with increasing temperature. However, undesirable changes in taste, color, and aroma may occur after heat treatment [92,94,95,96,97]. In addition, high temperatures may reduce the nutritional value and functionality of bioactive compounds found in fruits and fruit-based products [95,98,99,100].

Refrigerated storage can effectively slow the activation of the oxidases to stabilize the levels of antioxidants. After the pasteurization, storage temperature is an important factor that influences the qualities of fruit products during storage [101].

Due to their high moisture content, fruits are highly perishable; therefore, the use of other preservation methods is necessary to increase their shelf-life and reduce product wastage and environmental problems [26,102]. Drying treatments are used to reduce the water activity of fresh fruits in order to extend their shelf-life. In addition to whole fruit or dried fruit pieces, new products such as fruit powder are also being developed [103,104]. Different methods are used in the food industry to reduce the moisture content of fruits such as solar dryer, microwave heating, infrared irradiation, fluidized drying, or vacuum drying [105]. The system applied plays a key role in the final moisture content, quality, and shelf-life of the product obtained [106].

Freezing is a widely used long-term preservation method for fruits, where they retain attributes associated with freshness much better than other conventional preservation methods such as sterilization and drying [107,108]. Frozen fruits retain physical structure, nutritional and sensory attributes. In recent years, the processing of fruit pulp and juice has become an important economic activity for the production sector, as it adds cost-effective value to fruits, and minimizes losses [109]. Frozen fruit pulps are an important source of raw material and bioactive compounds and are being frequently used in the food industry for the production of other fruit-based products [110].

### 3.2. Innovative Methods: High Pressure Processing

Consumer demand for more natural products has led to an increase in the research on minimal preservation innovative techniques, such as High Pressure Processing (HPP), High Pressure Homogenization (HPH), and Pulsed Electric Fields (PEF) [111], among others. HPP can treat fruit products, previously packaged in flexible or rigid plastic bottles, at values ranging from 300 to 600 MPa, 0 °C to 90 °C. Usually, for commercial applications, it operates at room temperature (acting as a cold pasteurization technique). In HPP treatment, the pressure is applied uniformly, rapidly, independently of the shape, and according to Pascal’s principle. The products size results in fewer challenges when the production is scaled up [112]. Another advantage is that HPP can assist in reducing the number of ingredients by eliminating the need for certain additives for product stability and preservatives used for food safety and shelf-life extension [113]. Moreover, due to its limited effect on the covalent bonds of low molecular mass compounds such as vitamins, color, and flavor compounds, HPP treatment could preserve nutritional value and the sensory properties of fruits and vegetables. In addition, the primary structure of low-molecular-weight molecules (vitamins, amino acids, volatile compounds, pigments, etc.) is not affected, allowing a better retention of nutrients and sensory properties of foods. Nevertheless, the effect of HPP on vegetable products varies depending on processing conditions (pressure, hold time, and temperature) and food form (whole, pieces, juice, purée, mousse, or smoothie). Intrinsic factors such as pH or plant varieties also influence the process. Due to this, very often contradictory results can be found for the same matrix [32].

The largest commercial equipment can process up to 525 L per batch, which more than doubled between 1998 and 2014. Machines can have a vertical or horizontal operating design, working in a discontinuous or semicontinuous mode, being in-pack or in-bulk units. Nowadays, one industrial machine can operate continuously, allowing for packaging after processing and avoiding using exclusively plastic as the packaging material.

Over the past 30 years, HPP has evolved from the status of an emerging processing method to an industrially reliable technology. An expert survey with participants from industry, research institutions and government agencies highlighted that high pressure processing (HPP), microwave heating (MWH), and ultraviolet light (UV) have the greatest commercial availability and growth potential for the 2015–2025 period [114]. The Japanese were pioneers in commercializing HPP foods in the 1990s, followed by France and Spain. Now, the North American countries account for around 45% worldwide.

HPP is the innovative most used technique since the 2000s. In fact, its first commercialized application, placed in the market in 1990, was a variety of fruit jams. In the last years, fruits and their preparations, joined to vegetable products, are included in a food sector in which pressurizing techniques are mostly used, reaching a 25% share of the market for high pressure processed foods [115]. Moreover, the installed HPP machines in the world are increasing year by year, i.e., at the end of 2019, there were around 520 industrial HPP units running in the world. This figure is about twice than 5 years ago. Moreover, the performance, as represented by the number of cycles per hour, has also greatly improved, mostly because of a significant reduction in the total cycle time.

The main challenges of pressurization techniques are equipment price and production. The economics are more favorable with in-bulk machine processing liquid foods because the vessel has a 90% filling ratio compared to 45% on average for an in-pack machine loaded with bottles. Since the 2000s, the treatment cost at 600 MPa has been reduced more than three times. This is mainly due to technical improvements that decrease the processing cycle time. The use of HPP contributes to the overall product cost. Although the cost of HPP per kilogram of food processed is significantly higher (5–10 times higher) than that of a thermal treatment, this can be minimized with current use. The lowest operating costs are achieved with the largest equipment. Depreciation of the equipment is the most important factor in the operating costs, and the equipment cost per liter decreases with an increase in vessel volume [112].

Regarding legal requirements to commercialized HPP food products, in Canada there is no legal barriers as HPP is considered no longer a novel process; in USA, the HPP products have to comply with microbial requirements, but the U.S. Food and Drug Administration has officially approved HPP as a non-thermal pasteurization technology that can replace traditional pasteurization in the food industry [116]. In Europe, the procedure is more cumbersome, and it is not unified. European countries provided different answers to food companies that were trying to understand better the legal requirements for commercializing a pressurized food.

Finally, it is remarkable that HPP has the advantage of being well perceived by consumers relative to other novel food processing technologies, which greatly contributes to the success of the commercialized products processed by HPP [117].

## 4. Impact of HPP on Antioxidant Activity during Storage of Fruits and Fruit Products

HPP has been applied in numerous fruit derivatives and preparations with other ingredients such as vegetables or milk. This processing technology extends shelf-life compared to fresh and heat-treated fruit products. Most of the studies report the effects only after processing by pressurization. It should be interesting to observe how the antioxidant activity can change after storage. This revision is summarized in Table 1 (fruits, pulp, and purée fruits), Table 2 (fruit juices), and Table 3 (beverages and others fruit-based products), which show the main effects of these preparations on antioxidant activities.

### 4.1. Fruit, Pulp, and Purée

The content of nutrients in fruits is affected by factors such as cultivar, growing conditions, stage of maturity, and storage [118,119]. Processing of fruit to obtain derivate products such as purée and pulps is important because of its seasonality, since it prolongs its shelf-life. Fruit maturity is one of the most important factors associated with the quality of the final processed product; therefore, selecting the right maturity stages is critically important and processors need to be concerned with this. Although HPP of >200 MPa is usually used for food sterilization, mild HPP (<100 MPa) can potentially be used to modify the physiological status and nutritional value of the whole fruit or its processed products.

Mukhopadhyay et al. [120] investigated and evaluated the effects of HPP treatment (300–400–500 MPa, 5 min) at two temperatures (8 °C and 15 °C) applied to Cantaloupe melon purée on antioxidant capacity during 10 days of storage at 4 °C. ORAC assay in fresh purée showed values of 5.6 μmol Trolox equivalent/100 g fresh weight. No significant differences were found in the antioxidant capacity values between fresh and treated samples, and there were no differences in the antioxidant capacity values between treated samples in the different pressure and temperature ranges studied. Furthermore, the values of μmol Trolox equivalent/100 g fresh weight of the treated purée were very stable during 10 days of storage.

HPP treatment can be used as an alternative to thermal pasteurization of jujube pulp [121]. Treatment at 400 MPa or higher for >20 min contributed to higher ascorbic acid retention and increased total phenols, total flavonoids, and antioxidant capacity. Compared to heat treatment, HPP (600 MPa, 20 min) retained more nutrients and slightly reduced the antioxidant capacity of jujube pulp. With increasing pressure, ascorbic acid retention slightly increased to 92.9 and 93.3% in 500 MPa and 600 MPa samples, respectively. On day 40, the degradation rate of ascorbic acid in HPP-treated jujube samples was 11.9% (4 °C) and 32.1% (15 °C). However, the degradation rate of ascorbic acid in Thermal Processed (TP) samples stored at 4 °C and 15 °C was 60.2% and 67.3%, respectively. Additionally, samples stored at 4 °C retained more ascorbic acid than those stored at 15 °C, and HPP-treated samples retained more ascorbic acid than thermal treated samples. The antioxidant capacity of HPP-treated and TP samples decreased during storage at 4 °C and 15 °C. However, compared to TP, HPP resulted in greater antioxidant capacity in jujube pulp (42% vs. 28%).

Other authors [122] evaluated the effect of mild HPP (20−80 MPa/10 min) on the quality of the whole fresh mango during postharvest storage. Treated mangoes have a higher vitamin C retention. In addition, during the storage period (days 10–16) and in the samples treated with 40 MPa, they had the highest vitamin C content. The DPPH values of the extracts of different pressure treated mangoes during storage ranged from 1.90 to 3.19 g vitamin C equivalent/kg of fresh weight. As the mango developed, the HPP-treated samples showed higher values compared to the control. High correlations were found between DPPH activity with total flavonoid content and phenolic content, indicating that phenolic and flavonoid compounds are the main contributors to the antioxidant activity. A similar trend is observed between DPPH levels and total reducing capacity. The HPP-treated fruit generally possessed higher total reducing capacity compared to the control sample during the postharvest storage. Mangoes treated with 20 MPa showed higher antioxidant activity than that treated with 40, 60, and 80 MPa during storage. A higher content of antioxidants (phenolic compounds, flavonoids, and carotenoids) was also observed in the mango treated with 20 MPa during storage. The effect of different applied HPP conditions (400, 450, and 500 MPa for 2, 4, 8, and 16 min) and treatment temperatures of 25 and 55 °C on the content of total phenolics, total vitamin C, carotenoids and antioxidant activity has also been evaluated for preserving mango pulp [123]. Although changes during storage are not studied, it is evident that 550 MPa treatment combined with moderate temperature and processing times of 8 min achieve maximum retention of antioxidant compounds.

Peeled and diced ripe persimmon fruits were subjected to two different treatments of 200 MPa/3 min and 200 MPa/6 min at 25 °C. The HPP-treated samples and the control (untreated) sample were stored under refrigeration at 4 °C for 21 days (Table 1). HPP treatment resulted in a significant increase in carotenoid extractability (23–28%), but a decrease in total phenols (20%). During storage, a significant reduction in carotenoid and phenolic contents and in antioxidant capacity in all persimmon samples. On the other hand, the antioxidant activity decreased in the two pressurized samples until the end, showing values of 172.52 and 141.12 (µM Trolox eq/100 g fresh weight), respectively. In addition to carotenoids, the studied persimmon pulp has other constituents with antioxidant capacity such as vitamin C and polyphenols. A linear correlation was observed between total phenolic content and antioxidant activity. Thus, the degradation of phenolic compounds during storage in pressurized samples may explain the decrease in radical inhibition of these samples [124].

The application of moderate intensity pulsed electric fields (MIPEF) and HPP could be proposed as a strategy for producing and preserving plum purées with high antioxidant potential. Several authors [125] evaluated the effects on bioactive compounds of plums processed with HPP or moderate heat treatment and treated with and without MIPEFs during storage (40 days at 4 °C). The application of pulsed electric fields on plums slightly increased the levels of anthocyanins and the antioxidant activity of purées. The application of HPP increased the levels of bioactive compounds in purées, while the thermal treatment preserved better the color during storage. The application of MIPEFs to plums slightly increased the levels of some bioactive compounds such as anthocyanins (values from 6.5 to 7 mg/100 g fresh weight) and the antioxidant activity (values from 270.4 to 306.6 mg Trolox/100 g fresh weight) of plum purées. On the other hand, the addition of acid ascorbic during the manufacture was a deciding factor for the final quality of purées to preserve the color and the bioactive compounds content. In addition, the application of HPP as mild processing technology maintains the levels of bioactive compounds in purées.

Other researches [126] evaluated the impact of HPP (400 MPa, 5 min) and heat treatment (75 °C, 20 min) of cupped strawberries on total phenol content, total anthocyanins, and antioxidant capacity (DPPH and FRAP) during storage (45 days at 4 °C or 25 °C). All treated samples showed a reduction in the level of total phenols and anthocyanins of flesh during storage at 4 and 25 °C. In addition, there were higher total phenol and anthocyanin contents of flesh in samples stored at 4 °C and HPP-treated samples, compared to those stored at 25 °C and TP-treated ones, respectively. Moreover, HPP-treated samples stored at 4 °C showed the highest total phenols (805.8 mg GAE/100 g) and anthocyanins (19.4 mg/100 g) in all samples after a 45-day storage. Therefore, this result suggests that low storage temperature and HPP treatment inhibited the degradation of phenols and anthocyanins in cupped strawberry. The level of total phenols and anthocyanins was correlated with antioxidant capacity, which indicated that the reduction and increase of antioxidant capacity (DPPH and FRAP) in pulp during storage is mainly related to phenol and anthocyanin contents.

Marszałek et al. [127] determined the shelf-life of strawberry purée preserved with HPP and studied the rate of degradation of bioactive compounds (total polyphenols, anthocyanin pigments, phenolic acids, flavonols, vitamin C) during cold storage at 6 °C. A previous study [94] showed that the 500 MPa and 50 °C treatment was the most suitable for optimum microbiological quality and enzyme activity. Milder conditions may make the product equally valuable in terms of chemical quality but with a shorter shelf-life. The loss of L-ascorbic acid in the purée samples was high during the first two weeks of storage, 44% and 67% being quantified in the pasteurized and HPP-treated purée, respectively. No vitamin C was detected in the thermally pasteurized purée after 8 weeks of storage, whereas complete degradation was observed after 4 weeks of storage for the HPP-treated purée. On the other hand, the loss of anthocyanins in mash preserved in HPP during storage was 69% after 12 weeks, while only a 19% decrease was observed in pasteurized samples during this period. Another study [128] reported similar losses of anthocyanins in the HPP-treated (400 MPa, 5 min) strawberries stored at 4 °C for 45 days.
molecules-26-05265-t001_Table 1Table 1Effect of HPP treatment and storage on the antioxidant capacity of fruit purées and pulp.Fruit ProductTreatment ConditionsAntioxidant Method (Units)Main Effects after StorageReferencesCantaloupe puréeHPP-1: 300, 400 and 500 MPa, 5 min, 8 °CHPP-2: 300, 400 and 500 MPa, 5 min, 15 °CStorage: 10 days, 4 °CORAC (μmol Trolox/100 g fresh weight)The antioxidant capacity values of pressure treated purée were very stable during 10 days of storage at 4 °C.[120]Jujube pulpHPP: 400, 500 and 600 MPa, 20 min TP: 100 °C, 10 minStorage: 40 days, 4 °C or 15 °CAscorbic acid: Spectroscopy (mg AA/100 g of fresh weight)TPC: Folin-Ciocalteu (mg RE/100 g fresh weight)Total flavonoids: Spectroscopy (mg GAE/100 g fresh weight)DPPH (mg vitamin C equivalent/100 g fresh weightThe ascorbic acid content of HPP-treated samples stored at 4 °C and 15 °C decreased with increasing storage time.The antioxidant capacity of HPP-treated and TP samples decreased during storage at 4 °C and 15 °C. However, compared to TP, HPP resulted in greater antioxidant capacity in jujube pulp (42% vs. 28%).[121]Fresh MangoHPP: 20, 40, 60 and 80 MPa, 10 min, 20 °CStorage: 16 days, 13 °CDPPH (g vitamin C equivalent/kg fresh weight)ABTS (g vitamin C equivalent/kg fresh weight)Vitamin C, total phenolics, flavonoids and carotenoids: Spectroscopy (mg/kg fresh weight)Increased bioactive substances (vitamin C, total phenolics, flavonoids, and carotenoids) and antioxidant activities (DPPH and ABTS).Mangoes treated with 20 MPa show higher antioxidant activity than that treated with 40, 60 and 80 MPa during storage.[122]Persimmon piecesHPP: 200 MPa, 3 and 6 min, 25 °CStorage: 21 days, 4 °CTotal carotenoids: Spectroscopy (μg/100 g fresh weight)TPC: Folin-Ciocalteu (mg GAE/100 g fresh weight)DPPH (μmol Trolox/100 g fresh weight)During storage, antioxidant activity diminished in the two pressurized conditions.Total carotenoid content was not modified significantly up to 28 days in nontreated fruit.The degradation of phenolic compounds during storage in pressurized samples may explain the decrease in DPPH.[124]Plum puréeHPP: 600 MPa, 230 s, 25 °CTP: 75 °C, 30 sStorage: 40 days, 4 °CTotal anthocyanins: Spectroscopy (mg/100 g fresh weight)TPC: Folin-Ciocalteu (mg GAE/100 g fresh weight)ABTS (mg Trolox/100 g of fresh weight)The addition of ascorbic acid increased importantly the total antioxidant activity. At day 20 and 40 of storage, the purées with the highest total antioxidant activity and polyphenols content were those purées manufactured with ascorbic acid addition and processed by TP.[125]Cupped strawberryHPP: 400 MPa, 5 minTP: 75 °C, 20 min Storage: 45 days, 4 °C or 25 °CTPC: Folin-Ciocalteu (mg GAE/100 g of flesh)Total anthocyanins: Spectroscopy (mg/100 g of flesh)DPPH (mg Trolox/100 g flesh)FRAP (mg Trolox/100 g flesh)Reduction of antioxidant capacity in all samples (HPP and TP) during storage.Reduction in total phenolics, total anthocyanins and pulp antioxidant capacity, being more striking in the samples stored at 25 °C.Cupped strawberry stored at 4 °C showed higher contents of total phenols, total anthocyanins, and antioxidant capacity.[126]Strawberry puréeHPP: 300 and 500 MPa, 1, 5 and 15 min, 0 °C and 50 °C TP: 90 °C, 15 minStorage: 12 weeks, 6 °CTPC: Folin-Ciocalteu (mg GAE/100 g fresh weight)Phenolic acids and flavonols compounds: HPLC-DAD (mg/100 g fresh weight)Anthocyanin: Spectroscopy (mg/100 g fresh weight)Total vitamin C: Spectroscopy (mg AA/100 g fresh weight)No vitamin C was detected in the thermally pasteurized purée after 8 weeks of storage, while complete degradation was observed after 4 weeks of storage for HPP-treated purée.HPP-preserved purée had higher content of polyphenols and color parameters compared to TP purée.[127]HPP: High Pressure Processing; TP: Thermal Processing; TPC: Total phenol content; HPLC-DAD: High performance liquid chromatography-Diodes array detector; GAE: Gallic acid equivalent; AA: Ascorbic acid; RE: Rutin equivalent.

### 4.2. Fruit Juices

Some authors [128] studied the antioxidant activity and epicatechin content in apple juice, and found that, during the storage period, the content of epicatechin treated by TP and HPP decreased by 33% and 53%, respectively. The total antioxidant capacity of FRAP, scavenging rate of DPPH free radical were kept at 76%, 73% (TP), and 77%, 76% (HPP), respectively. The shelf-life of High Temperature Short Time (HTST) treated apple juice was longer than that treated by HPP (Table 2).

Apple juice was treated using HPP [129], and vitamin C content significantly decreased up to 30.9% after pressurization and at the end of storage. TPC increased after HPP, by a maximum of 6.1% at the highest pressure used. The antioxidant capacity measured by DPPH method was lower at longer storage. Nevertheless, ABTS test indicated an increase by 19.0% after pressurization and after 2 weeks, but decreased during the rest of storage. The antioxidant potential, in general, decreased with the storage time. The antioxidant activity was strongly correlated with the TPC and individual polyphenols.

For apple juice with Sabah snake grass leaves, after 36 days storage (4 °C) [130] total phenols increased from 4.54 g GAE/kg (untreated) to 4.91 in HPP-600 MPa samples. Regarding antioxidant activity, DPPH and FRAP decreased from 458 to 443 mmol/g and 330 to 326 mmol FeSO_4_/g, respectively. The total phenolic content and the antioxidant capacity (DPPH and FRAP) of the juice decreased significantly throughout storage for all treatments. All juices showed a decrease in citric acid during the first 15 days of storage. With increasing storage up to day 36, all juices exhibited an increase in citric acid regardless of the treatment applied.

In one study for aronia juice, some authors [131] observed that untreated juice had the greatest reduction (36%) in total polyphenols over the entire storage period (80 days). All the pressurized juices had significantly higher levels of phenolic compounds than their untreated samples upon storage. At the end of the storage period, the pressurized juices demonstrated ABTS and FRAP values higher by 14% and 5% as compared to the untreated juices. Regarding individual phenolic compounds, concentrations of cyanidin 3-xyloside in the HP-treated juices were still higher (by 33%, on average) than in the untreated juice after the entire storage. The pressurized juices stored for 40 and 60 days also had higher concentrations of cyanidin 3-arabinoside (58% and 10%, respectively) compared to the untreated juice.

In other work for pressurized sweet cherry juice [132], it was reported that TP increased TPC by 6%, HPP-400 had no effect while HPP-550 decreased them by 11%. During storage, phenols in control and TP samples decreased by 26% and 20%. TP had no effect on anthocyanins, while pressure treatments increased them by 8%. Anthocyanins decreased during storage, particularly in the control and P1 (decreasing 41%). All treatments had no effect on antioxidant activity until the 14th day; afterwards HPP showed the highest values.

Nayak et al. [133] found that the HPP samples showed higher phenolic content (495.71 mg GAE/kg) when compared with other samples in elephant apple juice. A significant difference was observed among the different juice samples. At the end of the storage period, the reduction in antioxidant compounds noted for both juices, pasteurized by heat juice and pressurized at 600 MPa, was approximately 38%, whereas juices untreated manifested 27.2% decrease in flavonoids concentration.

In jabuticaba juice [96] the storage (28 days at 7 °C) increased the total phenolics content, from 174 to 205 mg GAE/100 mL; total flavonoids from 39 to 56 mg QE/100 mL and monomeric anthocyanin from 1.86 to 2.50 mg/100 mL. The values for antioxidant activity ranged from 39 to 60% (ABTS antiradical activity), 32 to 34 (DPPH radical scavenging activity %) and 0.91 to 0.93 (reducing power). For reducing power there were decreases in all the samples during storage: the control, TP, HPP-200, HPP-400, and HPP-600 decreased by 25.4%, 10.8%, 24.2%, 23.8%, and 24.4%, respectively. Moreover, HPP samples showed lower browning degrees, compared with the TP treatment (Table 2).

According to the results of Zhao et al. in Korla pear juice [134], the antioxidant capacities of HPP and TP juices during storage were decreased with ascorbic acid and total phenols correspondently. First-order and linear model were used for fitting the decrease data of HPP and TP juices, respectively. Total phenol contents in HPP juice were higher than those in TP juice all through the storage; at the end of storage, total phenol contents in HPP and TP juices were 21.45 and 20.72 mg GAE/100 mL, respectively. A less and slower decrease in antioxidant capacity was observed in HPP juice compared to TP treatment. After 56 days of storage at 4°C, antioxidant capacity measured by DPPH method in HPP- and TP-treated juices were decreased 48.85% and 50.40%, respectively; and the antioxidant capacity measured by FRAP assay were decreased 8.57% and 11.36%, respectively.

For maoberry juice one study [135] reported that the retentions of ascorbic acid, phenols, and anthocyanins as well as antioxidant capacity (DPPH and FRAP assays) in pressurized juices were significantly higher compared to pasteurized juice, throughout the entire storage. In this study, high correlation coefficient of total anthocyanins and phenols versus antioxidant capacity was found.

The study of Zou et al. [136] in mulberry juice reported no significant difference in the antioxidant capacity among the untreated, the HPP-treated, and the TP treated juices immediately after the treatments. During 28 days of storage at 4 and 25 °C, the fluctuations of the antioxidant capacity in HPP-treated juices and TP-treated juices were similar to total phenols, which decreased after the initial 14 days of storage and then increased in the following days up to day 28. In High Pressure Carbon Dioxide (HPCD) samples the concentration of TPC significantly increased by 16%; no changes for HPP samples. After 28 days of storage at 4 and 25 °C, the concentration of total phenols in all the juices gained a significant increase. Compared with the untreated mulberry juice, the TP treated sample decreased only by approximately 4%. There was no change in the anthocyanin content in the HPP-treated mulberry juice. The anthocyanin content in all the mulberry juices decreased when increasing the storage time, and the decrease of anthocyanin content during storage at 25 °C was significantly larger than at 4 °C.

Other work in orange juice [137] found that the TPC of both TP and HPP orange juices decreased during storage, this behavior being more significant in TP juices, for which a decrease of about 25% was observed after 36 days. In HPP juices the highest content of anthocyanins was reported. The differences between the HPP and fresh orange juice are only significant after 22 days of storage, in opposition to that verified for TP juices, which present significantly lower anthocyanin content than fresh juice during the entire storage. Both TP and HPP treatments significantly decreased the total carotenoid content by 12% and 20%, respectively. HPP juice showed significantly higher carotenoid content than TP juice, during the 36 days of storage. Both treatments provoked a decrease in antioxidant activity measured by DPPH assay: this decrease was double in TP samples (26%) compared to HPP ones (13%).

Spira et al. [138] reported that pressurized and pasteurized juices exhibited the same strong reduction over time reducing around 70–80% of ascorbic acid levels in 90 days. Both HPP and TP orange juices exhibited the same behavior during shelf-life regarding TPC, with a slight decrease over time, and no significant differences between them. Antioxidant activity of pressurized, non-processed, and pasteurized orange juice using ABTS radical assay ranged from 294.7 to 302.3 mmol Trolox/100 mL, with no significant difference between both treatments. The correlation coefficient between TP and ascorbic acid was positive and strong for both juices. A positive and strong correlation was obtained between TPC and ABTS for HPP juice and a positive weak for pasteurized juice. ABTS and ascorbic acid presented a very strong and positive correlation for HPP juice and moderate and positive correlation for pasteurized juice (Table 2).

In an organic grape juice, Pasini et al. [139] did not find significant changes on antioxidant activity measured by ABTS assay during 6 months storage. The samples stored in polylactic acid (PLA) packaging showed lower PI (Polyphenol index) (−10%) than juice in polyethyleneterephthalate (PET) bottles, with a significant decrease during shelf-life. A total phenol reduction was tested for PET samples only at 6 months of storage. A significant increase from 44.8 to 60.8 mg RE/100 mL was observed for TP juices in the first 4 days of storage. From the 4th to 22nd day, the total flavonoid content in TP orange juice was not significantly affected. From the 22nd to the 29th day, the total flavonoid content decreased by about 19%. For all these indices, the HPP juices with PET packaging showed higher values compared to the control and the PLA counterparts. Besides, these last juices had a significant decrease during storage, whereas samples in PET bottles showed a strong decrease only at the 6th month.

In pitaya juice [140], after 45 days storage, a 10% decrease of total phenolic content was observed in HPP juice. Ascorbic acid (AA) in untreated juices showed a significant reduction (9%) after 15 days, while in HPP-treated juices the decrease reached up to 6% after 30 days (Table 2).

Gao et al. [141] found that total phenols in HPP-treated samples showed no significant difference compared with the control, but those in TP-treated samples decreased by 7.7% at the beginning of the storage. The content of total phenols treated by both HPP and TP showed no significant decline during 30 days, and the content of total phenols in HPP-treated samples was always higher than in TP-treated samples. After processing, HPP caused a 8.82% loss of ascorbic acid in grapefruit juice, while TP resulted in a 27.9% loss of ascorbic acid, indicating that thermal processing induced greater loss of ascorbic acid. During 30 days of refrigeration storage, ascorbic acid in HPP- and TP-treated samples showed a reduction of 21.1 and 22.4%. DPPH and FRAP was decreased by 5.0 and 14.3% in HPP-treated grapefruit juice and 5.3 and 12.8% in TP-treated grapefruit juice during the whole storage, respectively. The antioxidant capacity of HPP-treated samples was always higher than in TP-treated samples after 30 days, which was in accordance with the losses of total phenols and ascorbic acid by treatments of HPP and TP.

For a multifruit juice containing strawberry, apple, and lemon [142], the antioxidant activity measured by DPPH increased by 2.7% in HPP, and was maintained in heated (86 °C, 1 min) samples. Better maintenance of total phenols, total anthocyanins, ascorbic acid, and antioxidant capacity were observed in the HPP juice blend stored for 10 days at 4 °C, compared to both the US and HT treated samples.

In tomato juice [143] the fresh juices subjected to LPT (Low Pasteurization Temperature) and HPT (High Pasteurization Temperature) yielded 21% and 31% reductions in TPC, respectively. Among all the treated and stored (for 7 and 14 days) samples, the juice exposed to 600 MPa/15 min showed the highest TPC. The HPP treated juice showed a 15% reduction in TPC compared to the untreated sample. At the end of the storage, the juices exposed to HPP processing (400 and 600 MPa) had 21% and 10% lower TPC values than those of the fresh samples, respectively. The pressurized juices had 9% and 48% higher TPI values than those observed for the untreated and HPP-treated juices, respectively. The antioxidant capacity of the TP-treated juice was approximately 26% lower than the value noted for the untreated sample. A 15% drop in ABTS antioxidant capacity was noted for the TP-treated stored juices. The juice processed with 600 MPa for 15 min had the highest FRAP values of all the analyzed samples. The tested juices retained 95% of their reducing power at the end of the storage period. Similarly, the TP-treated juice also showed stable FRAP values during storage.

Chang et al. [144] reported a slight decrease in the total phenolic content of grape juice with increased storage time; however, this decrease was clearly observed in the TP samples on day 20. A decrease in the anthocyanin content was similar to the trend shown by the TPC during storage time. TP or HPP had no significant impact on the anthocyanin content. HPP significantly enhanced the ABTS value of grape juice from 10.64 mmol Trolox/L in the control group to 11.75 and 13.34 mmol Trolox/L in the HPP-300 and HPP-600 samples, respectively on day 0. The positive correlation between the antioxidant capacity and antioxidant content resulted in an increase in the total phenolic content on day 0 resulting from HPP, which accounts for the significant increase in the ABTS value. However, the prolonged storage period resulted in ABTS reduction from 7.7%, 20.4%, and 19.0% on day 20 in the HPP-300, HPP-600, and TP samples, respectively. Among these, the TP samples showed the most significant decrease, retaining only 71.4% of the ABTS value (Table 2).

In white grape juice concentrate, several authors [145] found that pressurization caused a decrease in total flavonoids and TPC: 8% and 25%, respectively, after 35 days of storage (4 °C). HPP caused a slight but significant decrease of both ORAC and DPPH values compared to the control samples.
molecules-26-05265-t002_Table 2Table 2Effect of HPP treatment and storage on the antioxidant capacity of juices.Juice ProductsTreatment ConditionsAntioxidant Method (Units)Main Effects after StorageReferencesApple juiceHPP: 400 MPa, 15 minTP: 98 °C, 50 sStorage: 70 daysDPPH (% inhibition)FRAP (% inhibition)Polyphenols: HPLC-DAD (μg epicatechin/mL)Antioxidant activity (DPPH) retention: 76% (TP) and 73% (HPP).Antioxidant activity (FRAP) retention 77% (TP) and 76% (HPP).Samples TP and HPP decreased polyphenol content by 33% and 53%, respectively.[128]Apple juiceHPP: 300 (3 pulses 5 min), 450 and 600 MPa, 5 minStorage: 90 days, 4 °CDPPH (μM Trolox)ABTS (μM Trolox)TPC: Folin-Ciocalteu (mg GAE/L)Ascorbic acid: HPLC-DAD (mg/L)Antioxidant activity (DPPH) increased up to 6.4% after HPP (600 MPa) and decreased with storage up to 21.5%. Antioxidant activity (ABTS) increased by 19.0% after HPP (600 MPa) and decreased with storage up to 36.0%. TPC increased after HPP, by a maximum of 6.1% at the highest pressure used. HPP treatment significantly decreased the total vitamin C content of apple juice between 2.5% and 30.9%.[129]Apple juice with Sabah snake grass leavesHPP: 300, 400 and 500 MPa, 5 min, 25 °CStorage: 36 days, 4 °CDPPH (mmol/g)FRAP (mmol FeSO_4_/g)TPC: Folin-Ciocalteu (g GAE/kg on a fresh weight basis)5.5–6.8% decrease in HPP samples.Almost unaltered: 1.2% decrease in HPP samples.Significantly lower amount of TPC than the untreated samples from 4–2%, but a decrease up to 10.8% in HPP samples after storage.[130]Aronia juiceHPP: 200, 400 and 600 MPa, 15 min, 26–38 °CStorage: 80 days, 4 °CABTS (mmol Trolox equivalents/mL)FRAP (mmol Fe^2+^ equivalents/mL)TPC: Folin-Ciocalteu (mg catechin/mL)During storage, the pressurized juices demonstrated ABTS andFRAP values higher by 14% and 5% as compared to the untreated juices.Juice HPP yielded a 12% drop (200 MPa) an 8% drop (600 MPa) yielded compared to the untreated samples and 36% drop at the end of storage.[131]Cherry juiceHPP: 400 and 550 MPa, 5 and 2 min, rtTP: 70 °C, 30 sStorage: 28 days, 4 °CDPPH (IC50, mL juice/mL DPPH)TPC: Folin-Ciocalteu (mg GAE/mL)Anthocyanins: Spectroscopy (mg CGE/L)TP samples had the lowest antioxidant activity after 28 daysA decline of about 26% for HPP samples and 20% for TP ones during storage. Both HPP treatments significantly increased the anthocyanins content (up to 15.5%) while TP was slightly decreasing them.[132]Elephant Apple juiceHPP: 600 MPa, 5 min, 35 °CTP: 80 °C, 60 sStorage: 60 days, 4 °CDPPH (µmol Trolox/g, % inhibition)TPC: Folin-Ciocalteu (mg GAE/kg)Flavonoids: Spectroscopy (mg quercetin/kg)In HPP samples, the antioxidant activity was maintained during storage but decreases in untreated samples (8.3%) and in TP samples (29.3%).TPC in control samples decreased 25.9% after 10 days storage and 25.9% for TP (60 days), while increasing up to 18.2% for HPP (60 days).A decrease was occurred in untreated samples after 10 days (27.2%) and also after TP (38.7) and HPP (37.7) during 60 days storage.[133]Jabuticaba juiceHPP: 200, 400 and 600 MPa, 5 min, rtTP: 90 °C, 60 sStorage: 28 days, 4 °CABTS (% antiradical activity)DPPH (% inhibition)Reducing power TPC: Folin-Ciocalteu (mg GAE/100 mL)Total flavonoids: Spectroscopy (mg quercetin equivalents/100 mL)Anthocyanins: Spectroscopy (mg/100 mL)Higher values for ABTS: 55–31% (600–400 MPa). No differences between HPP and TP for DPPH. The reductive power decreased significantly after storage. The control decreased significantly with increasing storage time, but the TPCs of HPP and TP samples remained constant. A significant decrease of total flavonoids with increasing storage time ranging from 27.9 to 43.3%. No significant changes for the samples treated at 400 and 600 MPa. Control, TP, and HPP-200 decreased anthocyanin content by 20.9%, 19.1, and 10.8%, respectively.[96]Korla pear juiceHPP: 500 MPa, 10 min, rtPretreatment: UFTP: 110 °C, 8.6 sStorage: 56 days, 4 °CDPPH (mmols Trolox/mL)FRAP (mmols Trolox/mL)TPC: Folin-Ciocalteu (mg GAE/100 mL)Ascorbic acid: HPLC (µg/mL)Antioxidant capacity (DPPH) in HPP- and TP-treated juices were decreased by 48.85% and 50.40%, respectively. Antioxidant capacity (FRAP) in HPP- and TP-treated juices were decreased by 8.57% and 11.36%, respectively. No significant difference of total phenols and ascorbic acid between UF pre-treated and UF-HPP juice, while they were significantly decreased by 4.73% and 13.43%, respectively, after UF-TP treatment.[134]Maoberry juiceHPP: 400 and 600 MPa, 10 min, 25–33 °CTP: 90 °C, 1 minStorage: 28 days, 4 °CDPPH (% of inhibition)FRAP (mmol Fe (II)/mL)TPC: Folin-Ciocalteu (mg GAE/100 mL)Ascorbic acid: HPLC-DAD (mg/100 mL)Anthocyanins: Spectroscopy (mg/L)Pressurized products retained higher antioxidant activity (75.1%) than pasteurized samples (65.0%) after total storage. A double reduction in FRAP values (49.4% against 25.7%) was obtained by TP compared to HPP treatment during storage. Total phenols were relatively stable under HPP, whilst a significant decrease of these compounds was found in TP at zero time (9.9%) and at 28 days (14.2%), respectively. Pressure levels had no effect on the loss of ascorbic acid. Anthocyanins decreased up to 10.5% in HPP and 24.3% in TP after 28 storage days.[135]Mulberry juiceHPP: 500 MPa, 5 minHPCD: 15 MPa, 55 °C, 10 minTP: 110 °C, 8.6 sStorage: 28 days, 4 °C and 25 °CDPPH (mmols Trolox/L juice)FRAP (mmols Trolox/L juice)TPC: Folin-Ciocalteu (mg GAE/L)Anthocyanins: Spectroscopy (mg/L)No differences among the untreated, the HPP-treated, and the TP treated juices immediately after the treatments. A slight increase in antioxidant capacity in the TP-treated juice.An increase in HPCD samples by 16%; no changes for HPP samples. After 28 days of storage at 4 and 25 °C, the concentration of TPC in all the juices gained a significant increase. After entire storage at 4 and 25 °C, the anthocyanins content in the TP-treated juice was decreased by 8% and 29%, respectively, and in the HPP-treated juice was decreased by 13% and 30%, respectively.[136]Orange juiceHPP: 550 MPa, 70 s, 18 °CTP: 70 °C, 70 sStorage: 36 days, 4 °CDPPH (mL/mg)TPC: Folin-Ciocalteu (mg GAE/100 mL juice)Flavonoids: Spectroscopy (mg rutin equivalent/100 mL)Anthocyanins: Spectroscopy (mg CGE/100 mL)Carotenoids: Spectroscopy (mg β-carotene equivalent/100 mL)Both treatments caused a decrease (26% TP and 13% HPP) in antioxidant activity.TPC decrease of 25.4% in TP samples and of 10.7% in HPP samples after 36 days. Increase of 26.3% after 4 days and 10.3% after 36 days for TP samples and decrease of 39.3% for HPP samples after storage.Similar decrease in HPP (14.5%) and TP (17.6%) samples after 36 days.A decrease in both treatments after 36 days: 22.3% (TP) and 26.6% (HPP).[137]Orange juiceHPP: 520 MPa, 6 min, 60 °CTP: 95 °C, 30 sStorage: 90 days, 4 °C *ABTS (µmol Trolox/100 mL juice)Ascorbic acid: Spectroscopy (mg AA/100 mL)TPC: Folin-Ciocalteu (mg GAE/100 mL juice)Antioxidant activity shows a slight increase during shelf-life and overall antioxidant activity for both juices maintained high values along shelf-life. Ascorbic acid reducing around 70–80% of ascorbic acid levels in 90 days. TPC showed a slight decrease over time, and no significant difference between TP and HPP.[138]Organic Grape juiceHPP: 600 MPa, 3 min, nrStorage: 6 months, 4 °C in PLA and PET bottlesABTS (mmol Trolox/L)Phenol index (PI): Spectroscopy (μg gallic acid/mL)Flavonol index (FI): Spectroscopy (μg galangin /mL)Anthocyanins: Spectroscopy (μg delphinidin chloride/mL)HPP do not cause significant changes on antioxidant activity.The PLA samples showed lower PI (−10%) than juice in PET bottles, with a significant decrease in shelf-life.The absorbances for FI and for anthocyanins showed the same trend reported for PI. The HPP juices with PET packaging showed the highest values. PLA juices had a significant decrease during storage, whereas samples in PET bottles showed a strong decrease at the 6^th^ month.[139]Pitaya juiceHPP: 550 and 600 MPa, 16 and 15 min, 20 °CStorage: 60 daysDPPH (mM Trolox/g in dry based) TPC: Folin-Ciocalteu (mg GAE/g in dry based)Betalains: Spectroscopy (mg/g in dry based)Antioxidant activity was not affected by HPP but had a 5% decrease after storage.Phenolic compounds were not affected by HPP but had a 10% decrease after storage.Betaxanthins showed no significant changes during storage (21.1 to 18.9 mg/g), while betacyanins showed a 28% reduction[140]Red grapefruit juiceHPP: 550 MPa, 10 min, 25 °CTP: 110 °C, 8.6 sStorage: 30 days, 4 °CDPPH (mmoles Trolox/L)FRAP (mmoles Trolox/L)TPC: Folin-Ciocalteu (mg GAE/100 g)Ascorbic acid: HPLC-UV-vis (mg/100 mL)Antioxidant activity decreased by 1.4% (DPPH) and 8.5 % (FRAP) after TP; no change with unprocessed samples during storage. DPPH and FRAP was decreased by 5.0 and 14.3% in HPP-treated grapefruit juice and 5.3 and 12.8% in TP-treated grapefruit juice during total storage, respectively. No differences in TPC in HPP juices, but TP-treated samples decreased by 7.7%After processing HPP caused 8.82% loss of AA, while TP resulted in 27% loss. During 30 days storage, AA in HPP- and TP-treated samples showed a reduction by 21.1 and 22.4%, respectively.[141]Strawberry, apple, lemon juiceHPP: 500 MPa, 15 min, 20 °CTP: 86 °C, 1 minUS: 376 W, 10 min, 35 °CStorage: 12 days, 4 °CDPPH (% inhibition)TPC: Folin-Ciocalteu (mg GAE/L juice) Anthocyanins: Spectroscopy (mg/L)Ascorbic acid: Spectroscopy (mg/100 mL)Increased by 2.7% in HPP, maintained in TP.Increased by 18% and 7% after HPP and US, respectively, and maintained by the TP.Maintained by HPP but decreased by 16% and 12% after US and HT, respectively. Ascorbic acid content in the juice blends were decreased by 9%, 11%, and 23% after HPP, US, and TP, respectively.[142]Tomato juiceHPP: 400 and 600 MPa, 15 min, 32–38 °C TP1: 74 °C, 2 min TP2: 90 °C, 1 min Storage: 14 days, 6 °CABTS (μmol Trolox/g fruit in dry weight)FRAP (μmol Trolox/g fruit in dry weight)TPC: Folin-Ciocalteu (mg GAE/g juice, dry based)TPI: HPLC (μg/100 g dry based)Greatest decrease 13% for TP1 and during storage and retention of 95% for HPP samples.Almost unaffected in HPP samples and a decrease up to 14% in HT samples.At the end of the storage period HPP samples increased between 4.6 and 22.1% of their polyphenol content and heated samples lost between 15.2 and 30.6%.HPP samples increased up to 6.6% during storage and decreased up to 50.6% in TP heated samples.[143]White grape juiceHPP: 300 and 600 MPa, 3 min, 20–29 °CTP: 90 °C, 1 minStorage: 20 days, 4 °CABTS (mmol Trolox/L)TPC: HPLC-DAD (mg/mL)Anthocyanins: Spectroscopy (mg/mL)HPP significantly enhanced the ABTS value of grape juice from 8 to 20% during storage. TPC showed the most significant decrease, retaining only 71.4% of the ABTS value. Slight decrease during storage and for both treatments. Better retention in HPP (97.4%) compared to TP (92.2%).[144]White grape juice concentrateHPP: 200, 300 and 400 MPa, 2 and 4 min, 20 °C; Storage: 35 days, 4 °CDPPH (µmol Trolox/100 mL)ORAC (µmol Trolox/100 mL)TPC: Spectroscopy (mg GAE/100 mL)Total Flavonoids: Folin-Ciocalteu (mg quercetin equivalent/100 mL)Individual phenols: HPLC-DAD (mg/L)HPP caused a slight but significant decrease of both ORAC and DPPH values. The retention of antioxidant activity significantly increased at HPP over 300 MPa/2 min, showing a positive effect of higher pressure.An increase of 8.9 and 37.1% in HPP samples at zero days and after 35 days, respectively.A decrease of 15.8% and 7.5% was observed in HPP samples at zero days and after 35 days, respectively.[145]*: refrigerated storage; HPP: High Pressure Processing; TP: Thermal Processing; UF: Ultrafiltration; HPCD: High Pressure Carbon Dioxide; PLA: Polylactic acid; PET: Polyethyleneterephthalate; US: Ultrasound; TPC: Total phenol content; HPLC-DAD: High performance liquid chromatography-Diodes array detector; GAE: Gallic acid equivalent; CGE: Cyanidin-3-glucoside equivalent; AA: Ascorbic acid.

Sánchez-Moreno et al. [146] have reviewed the effects of HPP on micronutrients. Briefly, it has been demonstrated that HPP techniques could retain more anthocyanins than thermal treatments, in various fruit juices [135,142,144] or do not change their content in other fruit preparations [96,136]. It has been shown that HPP could improve carotenoids content in comparison to thermal treatments, for two fruit preparations [147,148]. In general, HPP has been shown to have minor or no significant effect on micronutrients [149] and on total the carotenoid content within vegetable matrices [150]. In general, HPP treatments below 50 °C caused no degradation of ascorbic acid in fruit preparation products [134,135,141,142,151,152]. Regarding total flavonoids, there are contradictory results: Nayak et al. [133] reported the decrease of these compounds was less after pressurization, in comparison to heated samples; other work reported a similar decrease after storage in both HPP and thermal processing samples [96]; and Vieira et al. [137] showed that HPP caused a decrease of 39.3%. Studies focusing on other antioxidant vitamins, such as vitamin E (tocopherols and tocotrienols) and antioxidant minerals (selenium and zinc) present in fruit preparations are less abundant. da Silveira et al. [153] found no changes in vitamin E content after HPP treatments in acai (Euterpe oleracea) juice. Andrés et al. [154] studied milk and soy smoothies processed by HPP and found no changes in the zinc content after pressurization.

### 4.3. Beverages and Other Fruit-Based Products

#### 4.3.1. Syrup

Rinaldi et al. [155] used peach cubes previously treated with ohmic heating (ohm) to retain the structure of the samples to elaborate a syrup (Table 3). Afterwards, the samples were subjected to different pasteurization treatments: conventional pasteurization (ohm-DIM), ohmic heating (ohm-OHM), or high hydrostatic pressure (ohm-HPP), and were then stored at refrigerated temperature. The last treatment was the best to retain the overall quality, with only a reduction of 6% of the ascorbic acid against 16% and 22% for ohm-OHM and ohm-DIM, most likely for the thermal degradation. No significant variations were observed for total phenolic content (5% against the increase of 50% in the other two treatments, most likely due to the tissues’ damages in thermal treatments).

#### 4.3.2. Jam/Jelly

Strawberry jam prepared from two technologies (HPP and conventional thermal processing) and stored at 28 ± 2 °C in the presence of light were analyzed monthly for 3 months [156]. High pressure processing retained all antioxidant compounds, sensory characteristics, and microbiological quality in comparison with the thermal processed jam. The samples processed at HPP showed decreased contents in ascorbic acid (from 14.48 at 200 MPa pressure to 14.30 mg/100 g at 600 MPa), anthocyanin content (from 6.14 at 200 MPa pressure to 5.89 mg/100 g at 600 MPa), phenols (with the retention of 92.6, 91.7, and 90.8% at 200, 400, and 600 MPa pressure), total flavonoids (from 18.46 to 18.32 mg catechin eq./100 g in samples pressurized at 200 and 600 MPa, respectively) and antioxidant activity (from 93.08% at 200 MPa to 92.56% at 600 MPa). These values were found to decrease during storage in all pressure processed samples. However, the highest pressure applied to the sample was organoleptically preferred.

Shinwari and Rao [157] elaborated a reduced-sugar jam with mature sapodilla (*Manilkara zapota* cv. Kalipatti) fruits. Samples were treated with HPP (400 MPa for 10 min at 5 °C) and thermal processing (88 °C for 12 h). This fruit is an antioxidant-rich fruit, and the HPP-jams possessed a total phenol content about 25% higher than the thermally processed samples, and reported a reduction when the samples were stored more than three months at ambient temperature (Table 3).

#### 4.3.3. Beverages

Elbrhami [158] noticed a significant loss in vitamin C (from 8.22 to 5.94 mg/100 mL) and phenolic compound (from 139.14 to 95.85 mg GAE/100 mL) contents when HPP treatment at 600 MPa/120 s was applied to ‘horchata’ beverage, but the shelf-life was shown to be stable for up to 8 days at 4 °C (Table 3).

Swami Hulle and Srinivasa Rao [159] formulated a beverage with fresh litchi (*Litchi chinensis* cv. Bombai) fruits and aloe vera (*Aloe barbadensis* Miller) juice from leaves in a proportion of 85:15 (*v*/*v*). Samples were subjected to HPP treatment (600 MPa/15 min/56 °C) and TP (95 ± 1 °C for 10 min) and then stored in dark at three temperatures (4, 15, and 25 °C). The loss of ascorbic acid was higher for TP treated samples stored at 4 and 15 °C compared to HPP samples. The degradation for HPP-samples stored at 15 °C was up to 54% after 2 months and 58% for those stored at 25 °C after 30 days. With respect to the total phenolics for samples stored at 4 °C, HPP-samples presented contents ranging between 31 and 54 mg GAE/100 mL and in TP-samples ranging between 25 and 52 mg GAE/100 mL, but a decrease in phenolic content during storage in both treatments was recorded. The shelf-life of the HPP beverages was up to 110, 40, and 10 days for 4, 15, and 25 °C of storage-temperature.

Lemongrass-lime mixed beverages obtained at 200, 250, 300, or 400 MPa for 1 or 2 min and stored during 2 months at 4 °C were formulated [160], where ascorbic acid, total phenolic content, and antioxidant capacity (DPPH) were analyzed. The mixed beverage retained 100% of the original vitamin C content independently of the HPP treatment and time, most likely due to the low pH (about 2), but decreased drastically during storage (after 3 weeks decreased by 80.6%, and after 5 weeks, had lost 87% of their vitamin C compared with the fresh beverage (untreated at time zero)). Total phenolic content decreased slowly until 70% of the original beverage and stabilized at 4 weeks of storage. Antioxidant activity (DPPH) was found to correlate positively with both ascorbic acid and phenolic contents and dropped drastically (74%) after 3 weeks of storage and then slightly towards the end of the storage period; therefore 3–4 weeks was concluded as the self-life of this beverage.

Bansal et al. [99] formulated a beverage with whey, sweet lime, emblica juice, ginger juice, sugar, and salt. Three pressures (300, 400, and 500 MPa) for holding during different times (5, 10, and 15 min) at three temperatures (25, 35, and 45 °C) were assayed and 500 MPa, for 10 min at 25 °C were selected as optimal conditions. The samples were also subjected to thermal processing (90 °C for 1 min), and stored at 4 °C and analyzed during 120 days. At time 0, the bioactive compounds content was higher in HPP-beverages than in samples treated with thermal processing. After this time, gallic acid, chlorogenic acid, *p*-coumaric acid, rutin, naringin and hesperidin contents were reduced 2–3 times. The total phenolic content increased significantly from 549 mg GAE/L (fresh sample) to 558 mg/L after HPP treatment, and after the storage time (120 days) the 60.2% was retained (329 mg GAE/L).

A beverage formulated with papaya pulp and water (2:8, *w*/*v*) [147] was subjected to HPP treatment with four pressures (350–650 MPa) during 5 or 10 min and to TP at 110 °C during a short time (8.6 s). Total carotenoids, total phenols, and antioxidant activity were determined. No changes in carotenoids were found between the treatments and after storage, the retention was similar (about 70%). TP decreased significantly the content in polyphenols (13%) in comparison with HPP increased slightly (2%) the content, and after storage the total phenols contents decreased by 21.2% (HPP) and 25.8% TP). Consequently, DPPH and FRAP values decreased after TP treatment, and after 40 days of storage, the FRAP decreased by 37.2% (HPP) and 40.4% (TP), and the DPPH value decreased by 22.0% (HPP) and 35.8% (TP). The best combination of HPP variables were: 550 MPa during 5 min.

#### 4.3.4. Smoothies

A smoothie formulated with the extracted mango juice blended with the UHT skimmed milk (1: 2, *v*/*v*) was subjected to HPP treatments at 400, 500, and 600 MPa for 0–15 min or heat treatment (90 °C for 20 min) and then stored at 4 °C for 15 days [100]. The carotenoids content were analyzed on days 0, 3, 6, 9, 12, and 15 and decreased significantly and progressively with the storage time until a retention of 78.04%, 83.33%, and 83.72% in the mango smoothie samples treated by HT, HPP-500, and HPP-600, respectively, after 15 days of storage, most likely caused by geometric isomerization and susceptibility to oxidation of this family of compounds (Table 3).

Fernández et al. [151] formulated a smoothie with the following composition (weight): orange juice (59%), apples (15%), carrots (15%), beet leaves (6%), and beet stems (5%), which was treated with HPP (630 MPa for 6 min). Initially, HPP-treated samples presented higher values of ascorbic acid and total phenolics (about 7%) than in the control ones, and then a decrease took place with the storage time for both type of smoothies. This was more pronounced for vitamin C, most likely due to their oxidation to dehydroascorbic acid and the irreversibly conversion to 2,3-diketogulonic acid. HPP-smoothies showed initial higher values of the antioxidant capacities (5% in DPPH and 38.8% in FRAP method) in comparison with the control sample, then DPPH decreased gradually over time, but the FRAP value, after 14 days only remained about 98% of the initial FRAP activity. These results were also recorded in the same smoothie at 25 °C [161].

A multi-fruit elaborated [152] with apple juice (*Pyrus malus* v. Golden delicious), orange juice (*Citrus sinensis* v. Late Navel), strawberry (*Fragaria ananassa* v. Pajaro), whole apple, and banana (*Musa cavendishii* v. Pequeña enana) was subjected to three HPP treatments (350 and 450 MPa for 5 min and 600 MPa for 3 min) and thermal pasteurization (85 °C for 7 min). After 48 h under darkness, the antioxidant capacity is lower (higher IC50 and lower FRAP) in HPP-samples in comparison with TP-smoothies. Ascorbic acid content was low in treated smoothies, but not significantly different and no differences in the total contents of phenols and/or flavonoids were found with the HPP treatment after storage.

Formica-Oliveira et al. [162] in an orange smoothie with carrot and pumpkin also found a reduction of 31, 14, and 7% after 300, 400, and 500/600 MPa) of the initial total phenolic compounds content (712.1 mg chlorogenic acid equivalent/kg), but an increase of up to 1.6-fold after 7 days of storage, mainly in smoothies treated at 300 and 400 MPa.

Hurtado et al. [163,164] analyzed a red fruit-based smoothie composed by (g/kg) orange juice (200; pulp + juice), strawberry (200; pulp + juice), apple (160; squeezed apple juice), banana (100; squeezed banana juice), black grape (100; pulp + juice), blackberry (100; pulp + juice), gooseberry (70; pulp + juice), white grape (50; pulp + juice), and lime (20; squeezed lime juice) subjected to 350 MPa at 10 °C during 5 min or thermal processing, as well as analyzing microbiological, physico-chemical and sensory parameters every 7 days after 28 days stored alternately kept under artificial lighting (100 lx) (12 h) and darkness (12 h). The vitamin C present in smoothies was mainly L-dehydroascorbic acid and degradation during storage was higher in HPP-smoothies than in TP-samples (85 °C/7 min). The smoothies, independently of the treatment, showed a high phenolic content (about 850 mg GAE/L), high FRAP value (about μmol Fe^2+^/L), and DPPH value (about IC50 = 20) and decreased slightly during the 28 days of storage. Similar results were obtained by the same research group [165,166] in a fruity-vegetable smoothie elaborated with: blanched carrot *(Daucus carota*), apples (*Pyrus malus* golden delicious), citrus pectin solution 1%, 19.9% zucchini (*Cucurbita pepo*), pumpkin (*Cucurbita moschata* butternut), blanched leek (*Allium ampeloprasum porrum*), and salt, and by Picouet et al. [167] in a multi-fruit smoothie with apples (*Pyrus malus* v. Golden delicious), strawberries (*Fragaria ananassa* v. Pájaro), bananas (*Musa cavendishii* v. Pequeña enana), and oranges (*Citrus sinensis* v. Navel-late).

The multi-fruit milked smoothies elaborated by Andrés et al. [168] were composed by orange juice (*Citrus sinensis* cv Valencia Late), papaya juice (*Carica papaya* cv Maradol), melon juice (*Cucumis melo* L. Cantaloup), carrot purée (*Daucus carota* L. cv Nantes) and skimmed milk. They were treated with HPP technology (450 and 600 MPa/20 °C/3 min) and thermal processing (80 °C/3 min) and then refrigerated in storage for 45 days. The main carotenoids detected in the smoothies were lycopene, β-carotene, α-carotene, and δ-carotene; the first one, due to the papaya, showed the same content in samples pasteurized and non-treated and 16% higher in HPP-smoothie (450 MPa). Contents of β- and α-carotene were similar in all samples. Total carotenoid contents were slightly higher in HPP smoothies (10% at 450 MPa and 9% at 600 MPa), and the decrease was most important in the untreated (4.3%) and in the pasteurized samples (14.8%) after 45 days storage. Ascorbic acid decrease with thermal treatments, but to a lesser extent with HPP, and also decreased in all samples during storage. HPP smoothies showed significant increases in phenolic compounds, up by 6.6% (HPP-450) and 4.2% (HPP-600). FRAP and DPPH techniques showed that high correlation was found between FRAP and DPPH values with ascorbic acid and total phenolic content. This smoothie was also formulated with soymilk [148] and subjected to the same study, but the pressures applied were 550 and 650 MPa, coming to the same conclusions and correlations.
molecules-26-05265-t003_Table 3Table 3Effect of HPP treatment and storage on the antioxidant capacity of different beverages and fruit-based products.Fruit ProductTreatment ConditionsAntioxidant Method (Units)Main Effects after StorageReferencesPeach syrupPre-treated ohmic heating and HPP: 600 MPa, 3 minStorage: 4 °CTPC: Folin-Ciocalteu (mg catechin equivalents/kg)Ascorbic acid: HPLC-DAD (mg AA/kg)No significant increase of TPC (5%).Low reduction of ascorbic acid (6%).[155]Strawberry jamHPP: 200, 400 and 600 MPa, 30 min, 50 °CStorage: 3 months, room temperatureAscorbic acid: Spectroscopy (mg AA/100 g)Anthocyanin: Spectroscopy (mg anthocyanin/100 g)DPPH (%)TPC: Folin-Ciocalteu (mg gallic acid/100 g)Total flavonoids: Spectroscopy (mg catechin/100 g)Lower pressures: less reduction of the physico-chemical attributes but failed to yield proper gelling characteristics. Best sensory appeal: jam obtained at 600 MPa and was stable for a period of 3 months.[156]Sapodilla jamHPP: 400 MPa, 10 min, 27 °CStorage: 3 months, ambient temperatureTPC: Folin-Ciocalteu (mg GAE/100 g fruit)Reduction of TPC after storage.[157]Tiger nuts’ milkHPP: 400, 500 and 600 MPa, 90–120-180 s, 11 °CStorage: 8 days, 4 °CTPC: Folin-Ciocalteu (mg GAE/100 mL)DPPH (μmol Trolox/L)ORAC (mM Trolox)Loss in vitamin C content (8.22 to 5.93 mg/100 mL) of sample treated at 600 MPa/180 s. Decrease of TPC from 139.14 to 95.85 mg GAE/100 mL after 600 MPa/120 s. No modification of DPPH and ORAC.[158]Aloe vera-litchi mixed beverageHPP: 600 MPa, 15 min, 56 °CTP: 95 °C, 10 minStorage: 120 days, 4 °C; 60 days, 15 °C; 30 days, 25 °C in darkAscorbic acid: Spectroscopy (mg/100 mL)TPC: Folin-Ciocalteu (mg GAE/100 mL)Ascorbic acid content decreased at all the temperatures.Minimal changes in phenolic content. At 4 °C, the self-life was 100 days (HPP) and 80 days (TP).[159]Lemongrass-lime mixed beverageHPP: 200, 250, 300 and 400 MPa, 1–2 min, 25 °CStorage: 8 weeks, 4 °CAscorbic acid: Spectroscopy (mg/100 mL)TPC: Folin-Ciocalteu (μg GAE/mL)DPPH (μg Trolox/mL)HPP at 250 MPa ensured microbiological safety and no significant losses of vitamin C and phenolic compounds during the first 3–4 weeks.[160]Whey-based sweet lime beverageHPP: 500 MPa, 10 min, 25 °CTP: 90 °C, 60 sStorage 120 days, 4 °CTPC: Folin-Ciocalteu (mg GAE/L)DPPH (%)Self-life: 120 days (HPP) and 75 days (TP).60% (TPC) and 78% (DPPH) retention after storage.[99]Papaya beverageHPP: 350, 450, 550 and 650 MPa, 5 and 10 min, 20 °CTP: 110 °C, 8.6 sStorage: 40 days, 4 °CTotal carotenoids: Spectroscopy (mg/100 mL)TPC: Folin-Ciocalteu (mg GAE/100 mL)DPPH (Trolox)FRAP (Trolox)Similar retention (70%) of carotenoids with HPP or TP. Better preservation of TPC with HPP (79%) than with TP (74%).FRAP and DPPH decreased in HPP-samples by 37.2% and 22.0%, respectively.[147]Mango smoothieHPP: 400, 500 and 600 MPa, 0–15 min, 20 °CTP: 90 °C, 20 minStorage: 15 days, 4 °CTotal carotenoids: Spectroscopy (mg/100 mL)Carotenoid content decreased significantly in all treated mango smoothies.[100]Mixed fruit and vegetable smoothieHPP: 630 MPa, 6 min, 22 °CStorage: 28 days, 5 °CAscorbic acid: HPLC-DAD (mg AA/kg smoothie)TPC: Folin-Ciocalteu (mg GAE/kg smoothie)DPPH (μmol Trolox/kg smoothie)FRAP (μmol Trolox/kg smoothie)Increase in the initial values of nutritional quality indicators and decrease during refrigerated storage.[151]Mixed fruit and vegetable smoothieHPP: 630 MPa, 6 min, 22 °CStorage: 28 days, 5 °CTPC: Folin-Ciocalteu (mg GAE/100 g smoothie)DPPH (μmol Trolox/100 g smoothie)FRAP (μmol Trolox/100 g smoothie)Increase in the initial values of antioxidants indicators and decrease during room storage.[161]Orange vegetables smoothieHPP: 0, 300, 400, 500 and 600 MPa, 5 min, 23 °CStorage: 7 days, 5 °CTPC: Folin-Ciocalteu (mg chlorogenic acid/kg)TPC increased in all samples, mainly at 300–400 MPa.[162]Multifruit smoothieHPP: 350 and 450 MPa, 5 min, <25 °C, and 600 MPa, 3 min, <25 °CTP: 85 °C, 7 minStorage: 48 h, 4 °C under darknessDPPH (% inhibition)FRAP (μmol Fe^2+^/100 mL)Vitamin C: HPLC-DAD (mg/100 mL)TPC: Folin-Ciocalteu (mg GAE/100 mL)Total flavonoids: Spectroscopy (mg quercetin/100 mL)Decrease of FRAP and DPPH.Vitamin C, phenols and flavonoids were retained during storage.[152]Red fruit-based smoothieHPP: 350 MPa, 5 min, 10 °CTP: 85 °C, 7 minStorage: 28 days, 4 °C alternately (12 h/12 h) kept under artificial lighting and darknessDPPH (% inhibition)FRAP (μmol Fe^2+^/L)Vitamin C: HPLC-UV (mg/100 mL)TPC: Folin-Ciocalteu (mg GAE/100 mL)Degradation rate of vitamin C, total phenols, and antioxidant capacities during storage.[163,164]Multi-vegetables smoothie with appleHPP: 350 MPa, 5 min, 10 °CTP: 85 °C, 7 minStorage: 28 days, 4 °CFRAP (μmol Fe^2+^/100 mL)Vitamin C: HPLC-UV (mg/100 mL)TPC: Folin-Ciocalteu (mg GAE/100 mL)Total flavonoids: Spectroscopy (mg quercetin/100 mL)FRAP value and vitamin C content were very low and degraded during storage. Low levels of total phenols and flavonoids remained stable.[165,166]Multi-fruit smoothieHPP: 350 MPa, 5 min, 10 °CTP: 85 °C, 7 minStorage: 21 days, 4 °CDPPH (IC50)FRAP (μmol Fe^2+^/100 mL)Vitamin C: HPLC-UV (mg/100 mL)TPC: Folin-Ciocalteu (mg GAE/100 mL)Total flavonoids: Spectroscopy (mg quercetin/100 mL)Except for the flavonoids, which remained stable for up to 21 days of storage, all parameters decreased during storage.[167]Multi-fruit milk smoothieHPP: 450 and 600 MPa, 3 min, 20 °CTP: 80 °C, 3 minStorage: 45 days, 4 °CCarotenoids: HPLC-DAD (mg/100 mL)Ascorbic acid: HPLC-DAD (mg/100 mL)TPC: Folin-Ciocalteu (mg GAE/100 mL)FRAP (μmol Trolox/100 mL)DPPH (IC50)Ascorbic acid retention (95% and 92%) and decrease of total phenolics (12% and 11%), FRAP (45% and 61%) and DPPH (19% and 34%) for HPP-550 and HPP-650, respectively.[168]Multi-fruit soymilk smoothieHPP: 550 and 650 MPa, 3 min, 20 °CTP: 80 °C, 3 minStorage: 45 days, 4 °CCarotenoids: HPLC-DAD (mg/L)Ascorbic acid: HPLC-DAD (mg/L)TPC: Folin-Ciocalteu (mg GAE/L)FRAP (mg Trolox/L)DPPH (% scavenging)Decrease of total carotenoid content (4% for HPP-550 and 6% for HPP-650), ascorbic acid (43%), FRAP (26% at 550 MPa and 31% at 650 MPa) and DPPH.Increase (12%) in total phenolic content.[148]HPP: High Pressure Processing; TP: Thermal Processing; TPC: Total phenol content; HPLC-DAD: High performance liquid chromatography-Diodes array detector; GAE: Gallic acid equivalent; AA: Ascorbic acid; IC50: 50% radical inhibition.

## 5. Conclusions

Antioxidant activity varies widely during the storage of pressurized fruit and fruit products. Several factors may be involved, including fruit varieties and processed products (purées, juices, jams, beverages, etc.), processing parameters (pressure, temperature, pressure cycles and holding times), and storage temperature.

Literature describes a big variety of assays to determine the antioxidant properties and total phenolic, flavonoid, vitamin, carotene, or anthocyanin contents of fruit and fruit-based products. These methods require simple experimental conditions, and results are obtained after short times, but different units are used, and results are sometimes hardly comparable. Nevertheless, a positive correlation between bioactive compounds (anthocyanins, TPC, flavonoids, vitamin C and carotenoids) with antioxidant activity was observed. Regarding storage, in fruits and fruit preparations processed by HPP, most of the studies reflected a decrease during the storage period (higher at increasing storage temperature), but HPP better preserve the antioxidant compounds and antioxidant capacity than thermal treatments.

## Figures and Tables

**Figure 1 molecules-26-05265-f001:**
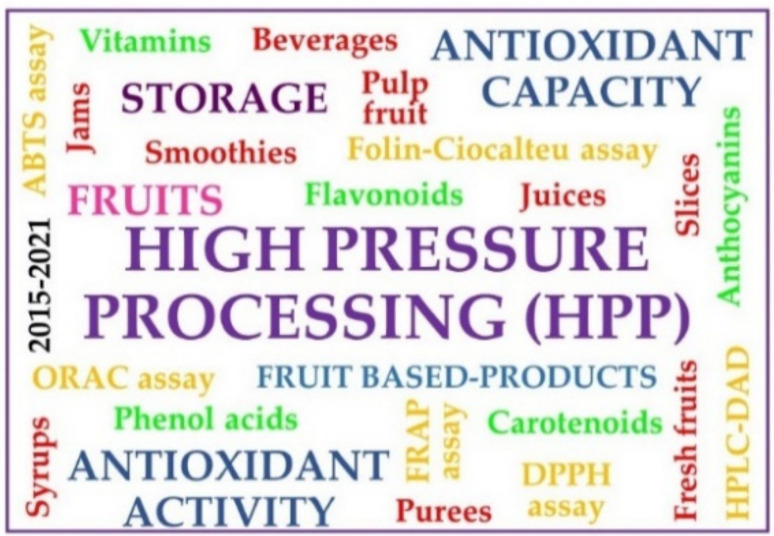
Target keywords of the review.

**Figure 2 molecules-26-05265-f002:**
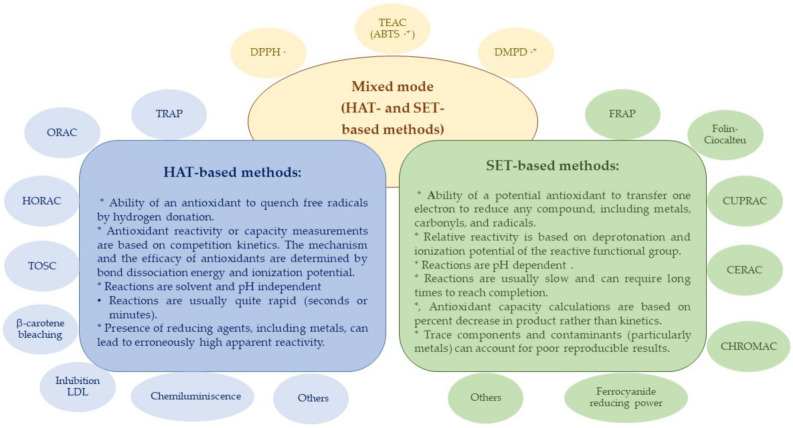
Characteristics of reaction mechanisms and main assays for the determination of antioxidant capacity.

**Figure 3 molecules-26-05265-f003:**
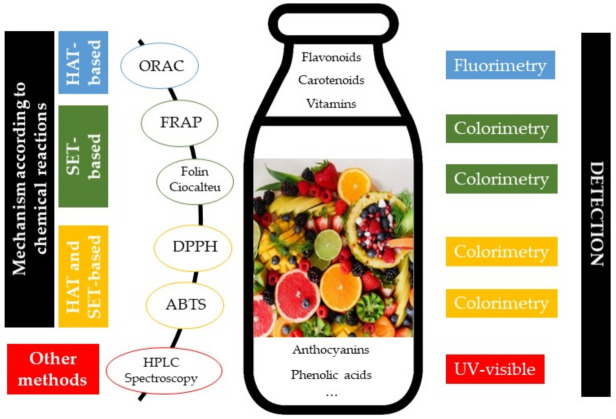
Chemical *in vitro* methods used for the determination of the antioxidant properties in fruits and fruit-based products.

## Data Availability

The data presented in this study are available on request from the corresponding author.

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
