# Peer review of "Impact of High-Pressure Processing on Antioxidant Activity during Storage of Fruits and Fruit Products: A Review"

_molecules, 2021, doi:10.3390/molecules26175265_

Round 1

Reviewer 1 Report

The authors made a great effort to classified fruits and fruits products, depending on processing parameters pressure, temperature, pressure cycles, and holding times during HPP, as well as storage temperature and their antioxidant activity and total phenolic, flavonoid, vitamins, etc. contents.

This work could be useful for scientists involved in the field of nutrition and food storage, generally.   

I recommended publication, after the addition of Briggs-Rauscher reaction as an antioxidant activity method (Helvetica Chimica Acta, 2001, 84(12):3533 – 3547, J Agric Food Chem, 2002, 18;50(26):7504-9, Archives of Biological Sciences 2019, 71, Issue 3, Pages: 425-434) in the Introduction section.

Author Response

Reviewer 1:

The authors made a great effort to classified fruits and fruits products, depending on processing parameters pressure, temperature, pressure cycles, and holding times during HPP, as well as storage temperature and their antioxidant activity and total phenolic, flavonoid, vitamins, etc. contents.

This work could be useful for scientists involved in the field of nutrition and food storage, generally.   

I recommended publication, after the addition of Briggs-Rauscher reaction as an antioxidant activity method (Helvetica Chimica Acta, 2001, 84(12):3533 – 3547, J Agric Food Chem, 2002, 18;50(26):7504-9, Archives of Biological Sciences 2019, 71, Issue 3, Pages: 425-434) in the Introduction section.

Response:

Thanks for your evaluation. Thank you for this proposed method, but cited assays in this review were the ones largely used as Folin-Ciocalteu, DPPH, etc. We only cited several reviews about the antioxidant capacity (concept, methods…). Moreover, the method based on Briggs-Rauscher reaction is not appearing in any of the cited references of our manuscript. Therefore, we consider it should be not mentioned in this article.

Reviewer 2 Report

Thank you for submitting the manuscript “Impact of high-pressure processing on antioxidant activity during storage of fruits and fruit products: a review” to Molecules. 

So, I even have few suggestions for this manuscript:
1) Use the attached PDF file to make corrections to the text. 
2) There is an inconsistency in the use of the comma. My suggestion is: 
Item A, Item B, or/and Item C
OR
Item A, Item B or/and Item C
Please correct all text. 

Author Response

Reviewer 2: (sent by a pdf file)

Response:

Thanks for the comments and suggestions (pdf file) to improve the manuscript. It was revised and updated to further improve mainly grammar and misprints, following reviewer 2 recommendations.

Lines 409-411: How the choice of manuscript that make up these tables (1-3) was made. Any manuscripts you found?

Response:

Previously, we checked the reviews based on HPP technology, and we not found any study focused in our subject. In Tables 1-3 appear all the works related to the effects of HPP, after storage, on antioxidant parameters of fruits and fruit-based products. Finally, we are decided as period reviewed the last five years from 2015 to the present.

Reviewer 3 Report

The manuscript is very appealing since it gathers a great collection of studies about how the high-pressure pretreatment of fruit products affects antioxidant activities and other bioactive compounds. In addition, it uses up-to-date bibliography, with some exceptions. However, the manuscript has some mistakes. Therefore, my recommendation is that the manuscript can be published in Molecules after minor revision.

Comments:

Keywords: “storage”, “fruits and fruit products”, “antioxidant activity”: are in the manuscript title. Please change these words to other keywords.

Line 39: “which make them” must be in singular “which makes them”

Line 45: please replace “consumption” with a synonym (intake) to avoid word repetition.

Line 65: there is missing an article “contributing to (the/a) circular economy”.

Line 75: please change “decades” with “years” or another word to avoid repetition (line 66).

Lines 74-75: Please remove “which can be” (…for specific health benefits attributed to the…).

Figure 1: I do not understand why this figure is here. The picture should be in the graphical abstract. It could be suitable for marketing purposes. Remove it please.

Line 87: Please correct “mecahnisms” for “mechanisms”.

Line 93: Please replace “such as” with “including”.

Line 98: Please reformulate the sentence. For instance: “…composition, etc. The two first mentioned are the most common, although they are not synonyms.”

Line 107: Please change the sentence “in food science and in the biomedical sciences” to “in food and biomedical sciences”.

Line 111: Please replace “big” with “large”.

Line 133: Please add “and” before “electron paramagnetic”

Line 168: Please correct “educing species” for “reducing species”

Line 172: I recommend writing the full name “gallic acid equivalent” before acronyms in parenthesis (GAE) since it is the first time it appears.

Line 204: please change “some authors” with “several authors”.

Lines 210-211: I recommend writing “Due to their high antioxidant capacity, they are important…” instead of “These bioactive compounds have high antioxidant capacity and are important for…”.

Line 219: Remove “But” at the beginning of the sentence “Main antioxidants…”.

Line 240: there is missing the full stop after “disorders”

Lines 244-248: Sentence is too large. I recommend reformulating the sentence to make them clearer and add punctuation where it is necessary.

Line 258: Remove “alpha-tocopherol” to avoid the word repetition (lines 257, 260) “It provides effective…”

Line 311: Please change “undesirable” to “unwanted” to avoid the word repetition (lines 308, 316).

Lines 357: Please correct the verb tense (vary) “…the effect of HPP on vegetable products varies depending on…”

Line 362: Please change the sentence “The machines can have vertical….” To “Machines can have a vertical...”

Line 363: Please correct “working in discontinuous “for “working in a discontinuous...”

Line 372: Please correct “accounted around” for “accounted for around”

Lines 429: Please always use the same word “pasteurisation” (UK), “pasteurization” (AM).

Lines 436, 439: acronyms “TP” and “TT” should appear in parenthesis after the full name.

Line 460: Please correct “Tª” for “temperature”.

Line 463: It is redundant the temperature of the treatments, so remove it (200 MPa/3 min and 200 MPa/6 min at a treatment temperature of 25 °C.)

Line 467: Please change “decrease” to “reduction” to avoid word repetition.

Lines 469, 473: pressurised (UK) Pressurized (AM)

Line 479: Please correct “of pulsed electric to plums” to “pulsed electric fields on”

Line 488: Please correct “technology that maintain the level” to “technology maintains the level”

Line 528: I recommend writing the full name “high temperature short time” before acronyms in parenthesis (HTST) since it is the first time it appears.

Lines 569-573: I recommend indicating the working pressure (MPa) in each result.

Line 577: Please correct “According results” to “According to the results”. Correct “capacity” to “capacities”.

Lines 587-591: I recommend reformulating the sentence to make it clearer and add punctuation where it is necessary.

Line 595-596: Please change “…TP-treated juices were similar to that of total phenols…” to “…TP-treated juices was similar to total phenols…”.

Line 597: “HPCD” the full name is in the abbreviation list. However, I recommend writing the full name before acronyms in parenthesis.

Line 609: Please change “in opposition with” to “in opposition to”

Line 616: remove “both”.

Lines 616, 618: Use plural of juice “juices exhibited”

Lines 616-618: Similar words in both sentences “both pressurized and pasteurized…”

Lines 619, 622: Please correct “significant difference” to “significant differences”.

Line 628: “PLA” the full name is in the abbreviation list. However, I recommend writing the full name before acronyms in parenthesis. Acronym “AA” should appear in parenthesis after the full name.

Line 640: Please correct “15 d” to “15 days”

Line 645: Please remove “that” in “higher than that in” (similar sentence to 652)

Line 652: Remove “that of” in “higher than that of TP”

Line 669: Please change “noted” to “observed”.

Lines 676-677: Please reformulate the sentence “A decrease in the anthocyanin content with increase in storage time, which is similar to the trend shown by the TPC was also observed”.

Line 683: Please change “resulted in decreases in ABTS” to “resulted in ABTS reduction”

Line 702: Please correct “studies focusing in” with “studies focusing on”

Lines 711-179: Sentences are too large. I recommend reformulating the sentences to make them clearer and add punctuation where it is necessary.

Lines 722-723: Please change “over a period of 3 months” to “for 3 months”.

Line 730: After parenthesis should be a full stop to start a new sentence.

Line 731: Before “however” should be a full stop to make another sentence. Moreover, I recommend reformulating the sentence like “However, the highest pressure applied to the sample was organoleptically preferred.”

Line 783: ¿PT or TP?

Line 791: Remove “together”. It is redundant.

Line 794: Reformulate the sentence like “The carotenoids content analyzed on days 0, 3, 6, 9, 12, and 15 decreased...”

Line 803: change “takes place” to “took place”

Line 805: extra dot at the end of the sentence (diketogulonic acid..)

Line 824: Please write “strawberry” instead of “straw-berry”

Line 836: “Citrus pectin” should not appear in capital letter

Line 842: Please write “smoothies elaborated…...were composed” instead of “smoothy elaborated……. was composed, because it is not only one smoothy.

Line 845: First sentence should end after “skimmed milk” (...skimmed milk. They were…)

Lines 849-853: Sentence is too large. I recommend reformulating the sentence to make it clearer and add punctuation where it is necessary.

Line 855: Please correct “increases of phenolic” to “increases in phenolic”.

Line 858: Please change “assayed” to “applied”.

Line 881: Please correct “the antioxidants compounds” to “antioxidant compound”

Line 882: Please correct the word “cpacity” (capacity)

Please check the abbreviation list and complete it. Some words are missing (HTST…).

Please check all tables, all information relating to a sample should appear together on the same page.

Authors use excessively the word “but”. They can employ other synonyms(though, although, however, nevertheless, nonetheless….).

Author Response

Response:

Thank you very much for your valuable comments and your detailed revision of the English. Most of them have been considered and the manuscript has been accordingly improved. 

All the answers to each comment are in the attached file.
